# Surface Water Purification using cellulose Paper Impregnated with Silver Nanoparticles

Shahad A. Raheem[1], Alaa H. Alfatlawi[2]

[1] College of Water Resources Engineering/ Al-Qassim Green University, Hilla/Iraq.
[2] College of Engineering / University of Babylon, Hilla/Iraq.
*Correspondence to*:  Shahad A. Raheem (E.mail:shahad.ak@wrec.uoqasim.edu.iq)

**Abstract.** The objective of this study is to prepare a cellulose paper that was impregnated with silver nanoparticles (AgNPs) for the purpose of water purification (Disinfection and filtration). AgNPs papers were prepared by chemical reduction of silver nitrate (AgNO$_3$) with various concentrations (0.005 M, 0.015 M, 0.03 M, and 0.05 M) using sodium borohydride (NaBH$_4$) as a reducing agent. Two ratios of NaBH$_4$/AgNO$_3$ of 2:1and 10:1 were used to show the effect of reduction on the formation and removal efficiencies of AgNPs. AgNPs papers were characterized using Scanning Electron Microscopy and Transmission Electron Microscopy. An acid digestion using HCL acid followed by analyzing the samples in Atomic Absorption Spectrometer was conducted to measure the silver concentration in AgNPs papers. TEM images showed that the silver nanoparticles size in the papers varied from 1.3 to 75 nm.

Water samples, after filtration through AgNPs papers, were analyzed using ASS to measure the silver concentration in the effluent water. AgNPs paper antibacterial efficiency ranged (99 % to 100 %) for both reduction ratios. The average silver content in the effluent water for the three replicates ranged from 0 to 0.082 mg/L which meets the United States- Environmental Protection Agency (US-EPA) guideline for drinking water of less than 0.1 mg/L. Turbidity tests showed that these papers can be usefully used as a point of use filters as the turbidity reduced to less than 1 NTU.

## 1. Introduction

Water is the common medium for many pathogens because it contains several bacteria, viruses, etc. The removal and inactivation of pathogenic microorganisms are the last step in the treatment of drinking water [Phong *et al*., 20009]. Although disinfection methods currently used in drinking water treatment can effectively control microbial pathogens, researches in the past few decades have revealed a dilemma between effective disinfection and formation of harmful disinfection byproducts (DBPs) [Li *et al.* ,2008]. When chlorine comes in contact with natural organic matter (NOM), carcinogenic compounds such as trihalomethanes (THMs) and haloacetic acids (HAAs) can be formed [Lalley *et al.,* 2014].

Nanotechnology and its application is one of the rapidly developing sciences [Hossain *et al.,* 2013]. It is an important field of modern research dealing with design, synthesis, and manipulation of particles structure ranging from approximately 1-100 nm [Korbekandi & Iravani, 2012]. Among the most promising nanomaterials with antibacterial properties are metallic nanoparticles, which exhibit increased chemical activity due to their large surface to volume ratios and crystallographic surface structure [Savage & Diallo, 2005]. Silver nanoparticles have proved to be most effective as it has good antimicrobial efficacy against bacteria, viruses and other eukaryotic micro-organisms [Rai *et al*., 2009]. The investigation of enhanced disinfection through the use of silver nanoparticles (AgNPs) surface immobilization has been continually explored. From silver doped

hydroxyapatite coatings for reduced infection rate of implanted biomedical devices [Bai *et al.,* 2012], to silver impregnated ceramic filters for point of use treatment in rural Guatemala [Kallman *et al.*, 2010], AgNP coated surfaces have displayed a wide range of potential applications.

These commonly encountered materials could be beneficial in maintaining bacteria-free water which is being stored or transported. Graphene, activated carbon, and nepheline films have also been studied for AgNP immobilized antibacterial surfaces [Lalley *et al.,* 2014].

Cellulose materials serve as a good material for embedding metal nanoparticles due to their ability for metal ion absorption. Metal cations have an affinity for anionic carboxylic acid groups in paper, the porosity of the base paper allows microorganisms to come into contact with the biocide, but attachment to the fiber surfaces limits the level of silver in the effluent water. The large pore size in the paper allows for reasonably rapid flow by gravity without the need for pressure or suction [Dankovich and Gray, 2011].

The main objective of this study is to prepare disinfection material that has high antibacterial efficiency, few side effects, and can be readily used for water treatment. This study consider the use of nanotechnology, specifically the use of silver nanoparticles (AgNP) in water disinfection purposes (inactivating of Escherichia Coli, Staphylococcus aureus, Enterococcus faecalis, Klebsiella Pneumoniae and Enterobacter Aerogenes).

## 2. Materials and Methods

### 2.1 Sampling

The study area was Shatt Al-Hilla River at Al-Hilla city/Iraq from which the samples were taken during the period (2018 – 2019), because this river is the main source of water in the city and to investigate the possibility of AgNPs papers ability to purify this water. It's characteristics are presented in (Table 1). A sample of 500 ml of water was grabbed and kept in precleaned plastic bottles. The samples were analyzed immediately to prevent any change in their quality that may occur.

**Table 1: Characteristics of raw water samples**

| Property | Value |
|---|---|
| Turbidity, NTU | 13.5 |
| pH | 8.4 |
| Total Dissolved Solids (TDS), mg/L | 921 |
| Temperature, ℃ | 18.4 |
| **Bacteriological analysis** | |
| Escherichia Coli, CFU/ml | 3300 |
| Staphylococcus Aureus, CFU/ml | 7750 |
| Enterococcus Faecalis, CFU/ml | 47100 |
| Enterobacter Aerogenes & Klebsiella Pneumoniae, CFU/ml | 3600 |
| Proteus mirabilis, CFU/ml | 150 |

### 2.2 Preparation of AgNPs papers

A (10 cm * 10 cm * 0.8 mm) off-white paper, 100% alpha cellulose Whatman Gel Blot GB005 was used to be embedded with silver nanoparticles. AgNPs papers were prepared by in situ reduction of $AgNO_3$ with various

concentrations (0.005 M, 0.015 M, 0.03 M and 0.05 M) and two reduction ratios of 2:1 and 10:1, to show the effect of increasing the concentration of the reducing agent on the formation of AgNPs. Each paper was soaked in 40 ml of AgNO3 solution for 30 minutes, then it was washed with ethanol for 1 minute to remove the excess Ag ions which were not adsorbed by the paper. To form AgNPs, the paper was placed in 40 ml of $NaBH_4$ solution for 1 hr. After that, the paper was soaked in de-ionized water for 30 minutes. Then the paper was dried

in the oven at 60 ℃ for 2.5 hrs. (Figure 1) shows the papers before and after embedding with AgNPs.

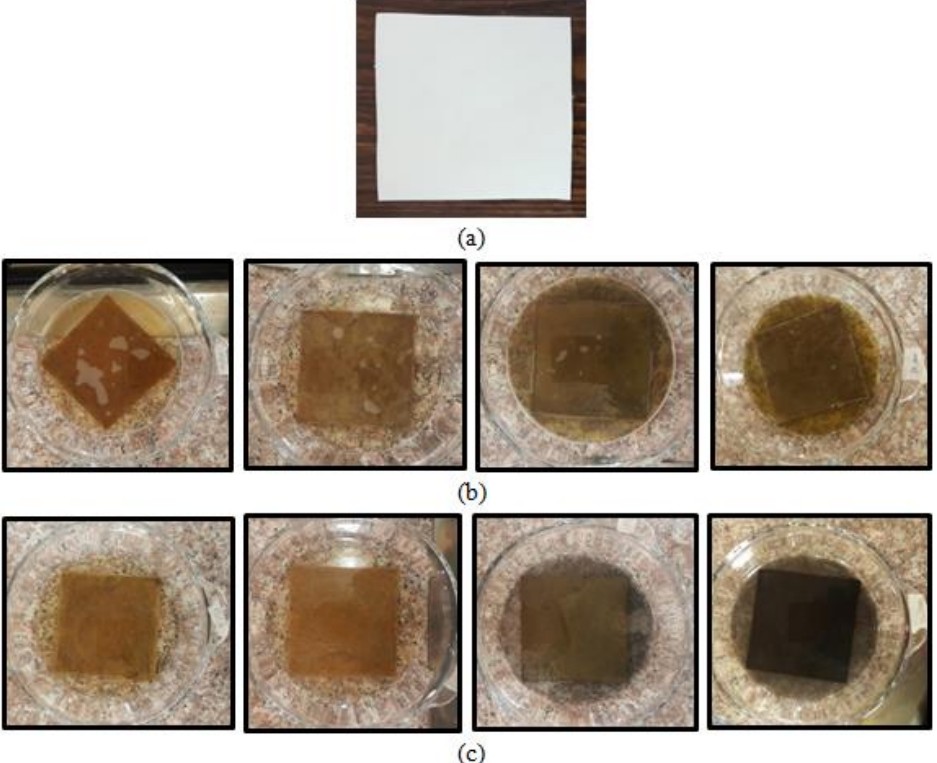

Figure 1: Cellulose papers: (a) Before being impregnated with AgNPs. (b) During preparation of AgNPs with $NaBH_4/AgNO_3$ ratio of 2:1. (c) During preparation of AgNPs with $NaBH_4/AgNO_3$ ratio of 10:1.

### 2.3 Characterization

The synthesized AgNPs papers were characterized by Scanning Electron Microscopy (SEM), type Quanta 450 available at the University of Babylon/ College of Pharmacy and Transmission Electron Microscopy (TEM) available at Al-Nahrain University/ College of Medicine.

### 2.4 Acid Digestion

To determine the amount of dissolved silver was analyzed in the AgNPs paper, an acid digestion of the paper

was performed and then analyzes the amount of dissolved silver with an Atomic Absorption Spectrometer (AAS, type AA320N) available at the University of Babylon/ College of Material Engineering. Approximately a 100 mg of the dried AgNPs paper was reacted with 5 ml of nitric acid ($HNO_3$) and 5 ml of water. The mixture

was boiled until the paper was disintegrated. 5 ml of 30% hydrogen peroxide ($H_2O_2$) was added to the mixture to assist in the complete oxidation of the organic matter to release additional metals into the solution. The mixture was boiled again and left to be cooled, then filtered through Whatman filter paper (Grade 41) with diameter of 15 cm and then diluted by adding a 100 ml of water. The diluted mixture was tested for silver content using an AAS.

**2.5 Microbiological Test**

Urinary Tract Infections (UTI) chromogenic agar was prepared by suspending 47.5 gm of the medium in 1 L of distilled water. The mixture was mixed well and dissolved by heating with frequent agitation. Then it was boiled for 1 min until complete dissolution. The media was sterilized by placing it in an autoclave at 121 ℃ for 15 min then it was cooled to 45-50 ℃ and mixed well and dispensed into plates and left to be solidified. The dehydrated medium was homogeneous, free-flowing and beige in color.

The samples were cultured using serial dilutions method, 1 ml of the sample was diluted in 9 ml of distilled water (1:10 dilution). this process repeated until 1:100000 dilution. 0.1 ml of each dilution was spread over a media plate and then the plates were incubated in 37 ℃ for 48 hrs in an incubator (LIB-030M). the colonies were counted by eye and colony forming units per milliliter were calculated using the following equation: [Hameed et al., 2015]

$$CFU/ml = \frac{No. of\ colonies * Dilution\ Factor}{Volume\ of\ the\ sample\ plated\ (ml)} \ \dots\dots\dots\dots \qquad 1$$

### 3. Results and Discussion

**3.1 Paper Characterization**

The AgNPs papers were characterized by SEM and TEM. (Figure2) represents the images obtained by SEM to show the presence of AgNPs in paper fibers and the images obtained by TEM to determine the particles sizes of AgNPs. (Table 2) represents the particles sizes of AgNPs obtained by TEM test.

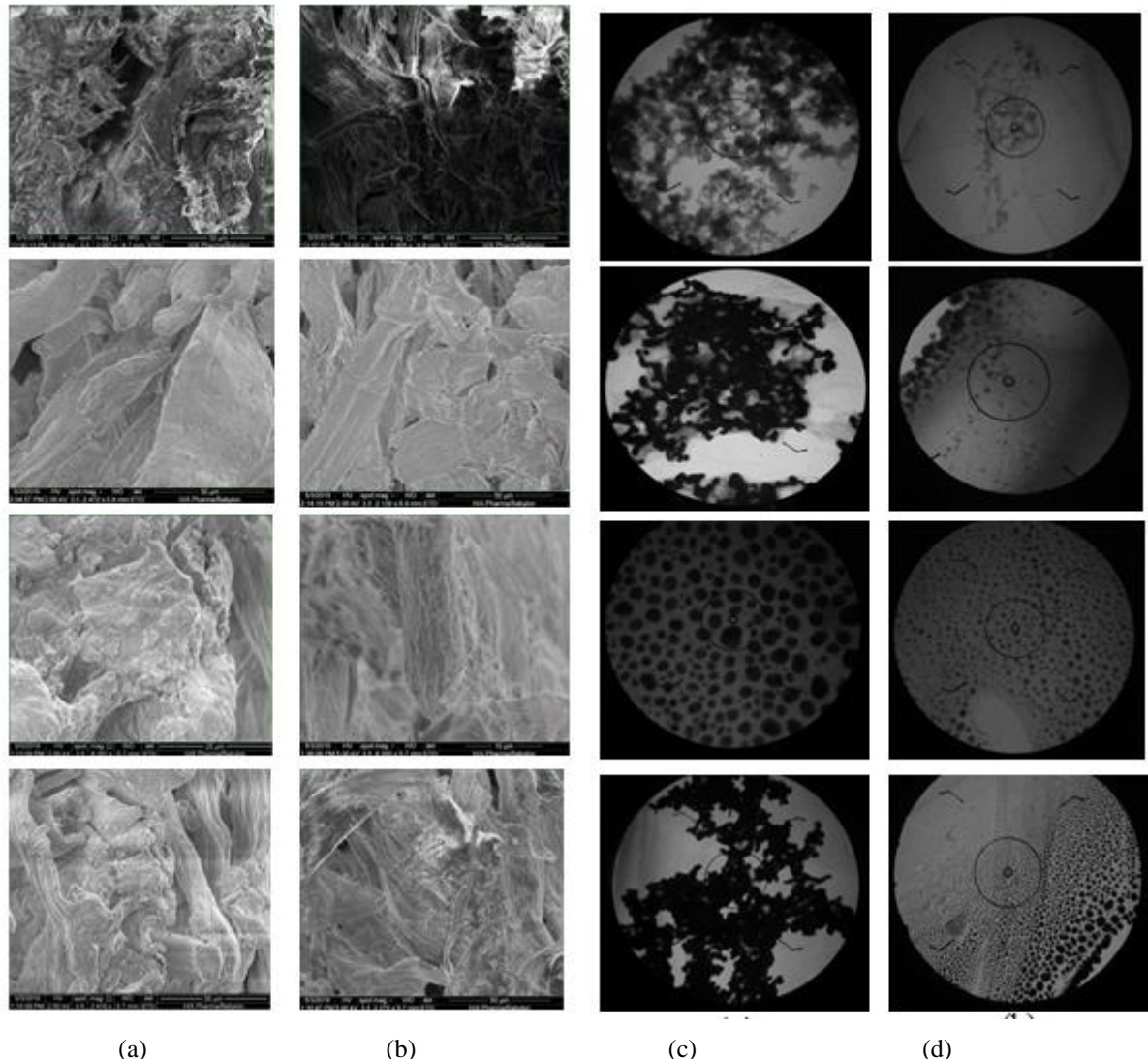

(a)          (b)          (c)          (d)

**Figure 2: (a) Images obtained from SEM using 2:1 NaBH$_4$/AgNO$_3$ ratio. (b) Images obtained from SEM using 10:1 NaBH$_4$/AgNO$_3$ ratio. (c) TEM images using 2:1 NaBH$_4$/AgNO$_3$ ratio. (d) TEM images using 10:1 NaBH$_4$/AgNO$_3$ ratio**

**Table 2: The particles sizes of AgNPs obtained by TEM test**

| AgNO$_3$ concentration, M | Nanoparticle Size Range ,nm | |
|---|---|---|
| | 2:1 NaBH$_4$/AgNO$_3$ ratio | 10:1 NaBH$_4$/AgNO$_3$ ratio |
| 0.005 | 6.86 - 75 | 2.028 – 39.395 |
| 0.015 | 3.399 – 42.521 | 1.333 – 39.643 |
| 0.03 | 3.064 - 50.311 | 1.314 – 23.431 |
| 0.05 | 2 – 21.84 | 0.943 – 20.044 |

TEM images and results presented in Table 2 showed that an excess of sodium borohydride reductant (10:1 ratio of sodium borohydride to silver nitrate) resulted in more uniform and smaller nanoparticles. This can be due to

the increased speed of reduction with the increment in the reducing agent [Catalina Quintero *et al.*, 2019]. These results agree with the findings of the previous studies concerning this subject.

**3.2 Acid Digestion**

Acid digestion was performed to determine the silver content of the paper. The results were obtained by using (AAS, type AA320N). Table 3 shows the results of the AAS test.

**Table 3: AAS test showing the silver content of each paper**

| AgNO3 concentration, M | Silver content, mg Ag/g of dried paper | |
|---|---|---|
| | 2:1 $NaBH_4/AgNO_3$ ratio | 10:1 $NaBH_4/AgNO_3$ ratio |
| 0.005 | 3.958 | 4.343 |
| 0.015 | 4.625 | 4.698 |
| 0.03 | 7.007 | 7.911 |
| 0.05 | 7.867 | 8.769 |

The acid digestion of AgNPs papers showed silver content ranging from 3.9 to 8.7 mg Ag per dry gram of paper. The increase in silver content of the paper correlates with the increase in precursor silver ion concentration of the solution in which the papers were soaked, prior to reduction [Dankovich Theresa A. and Gray Derek G, 2011]. For the same concentration of $AgNO_3$, the $NaBH_4/AgNO_3$ ratio of 10:1 resulted in more silver content than 2:1 ratio. These results agree with the findings of the previous studies concerning this subject.

**3.3 Removal Efficiencies of Bacteria**

Figures 3 and 4 show the effect of the silver content in the AgNPs paper on the removal efficiency of different types of bacteria with a $NaBH_4/AgNO_3$ ratio of 2:1 and 10:1 respectively.

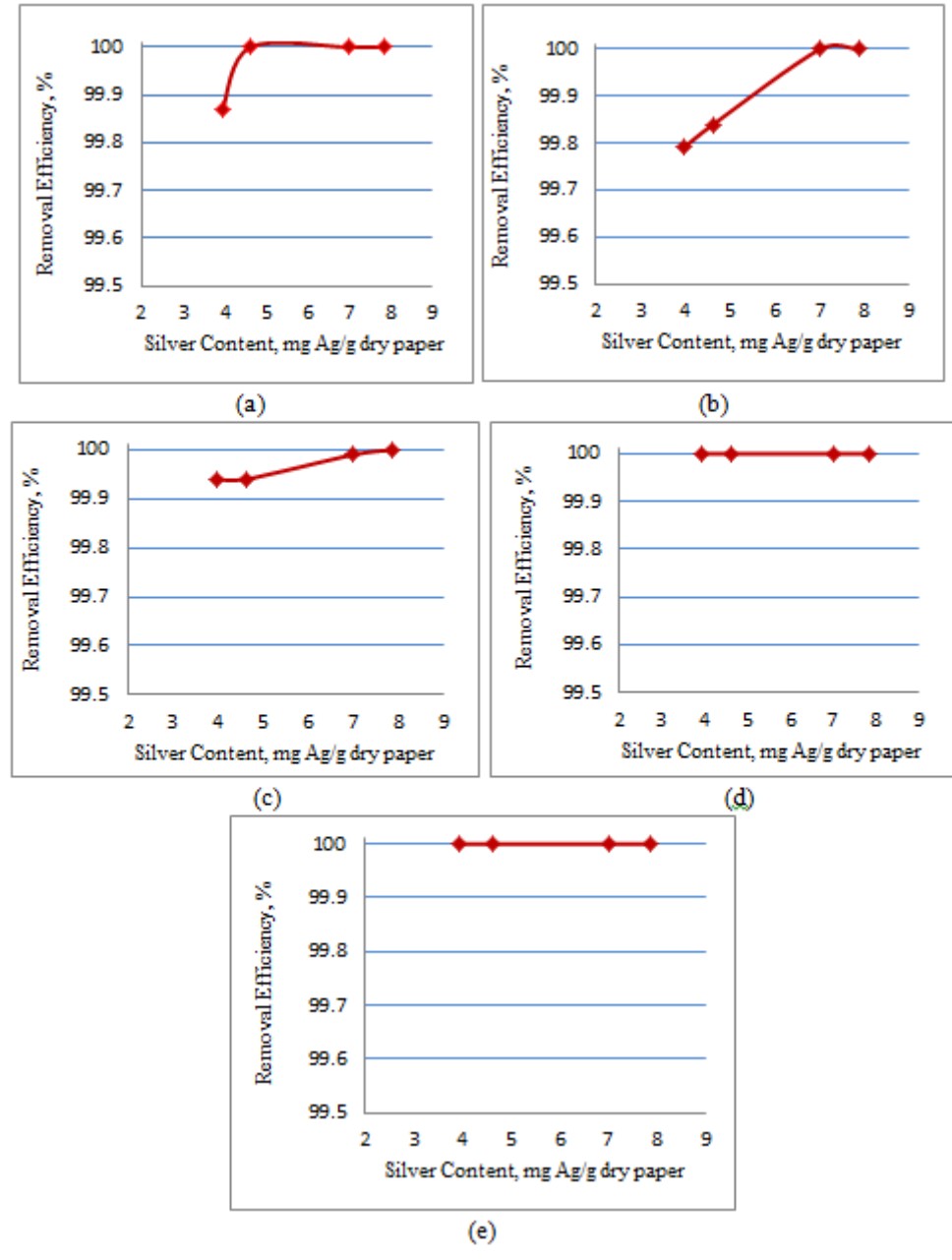

**Figure 3: The removal efficiency of different types of bacteria with NaBH$_4$/AgNO$_3$ ratio of 2:1: a: E. Coli. b: Staphylococcus Aureus. c: Enterococcus Faecalis. d: Enterobacter Aerogenes & Klebsiella Pneumoniae. e: Proteus mirabilis.**

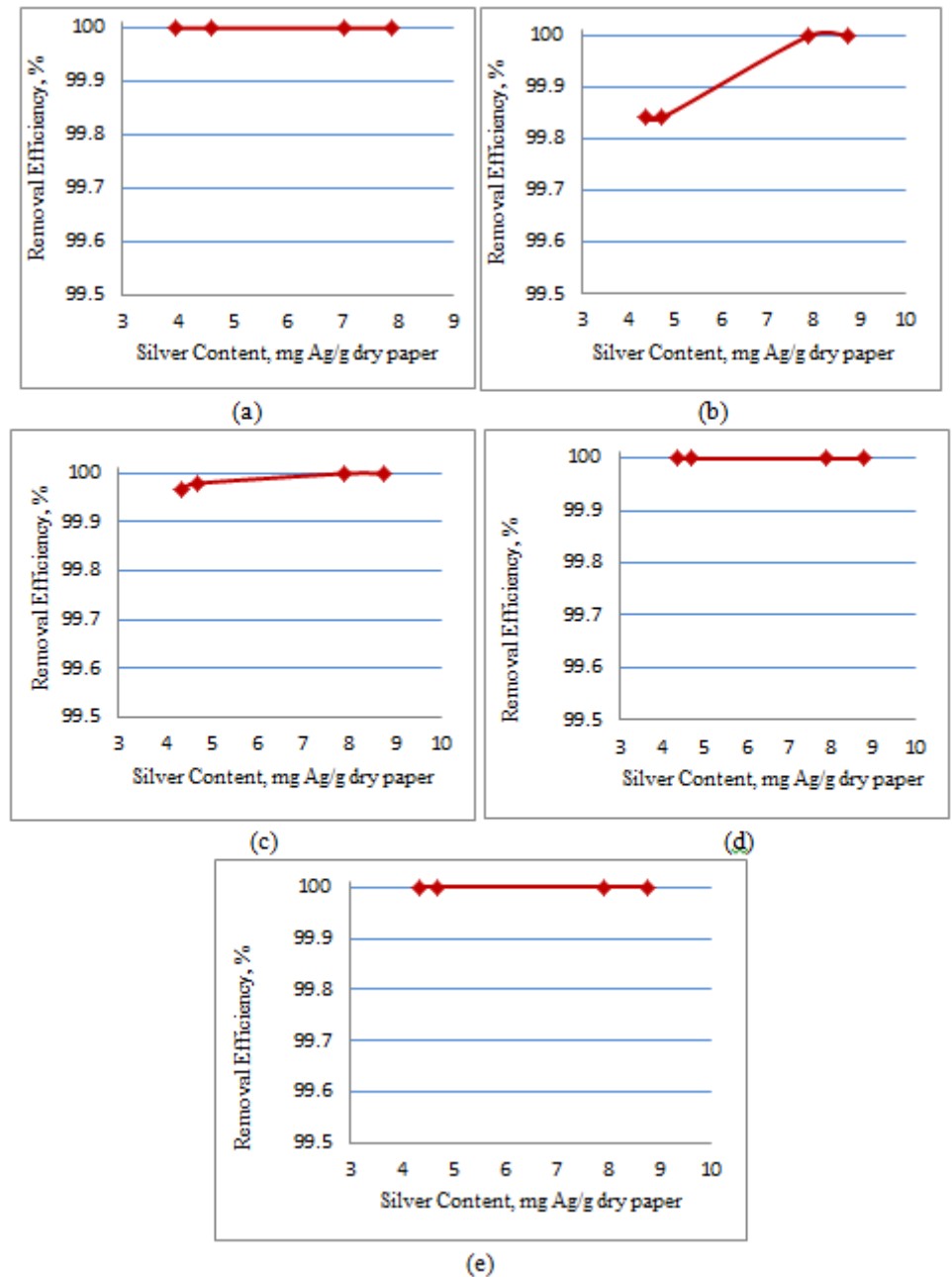

**Figure 4: The removal efficiency of different types of bacteria with NaBH₄/AgNO₃ ratio of 10:1: a: E. Coli. b: Staphylococcus Aureus. c: Enterococcus Faecalis. d: Enterobacter Aerogenes & Klebsiella Pneumoniae. e: Proteus mirabilis.**

As shown in Figures 3 and 4, the Minimal Inhibitory Concentration (MIC), which is the lowest concentration of
140 silver nanoparticles needed to inactivate the bacteria, for E. coli for all three filtration times was 4.62 mg Ag/g dry paper for a NaBH₄/AgNO₃ ratio of 2:1, while for 10:1, the MIC for complete inactivation of E. coli was 4.34 mg Ag/g dry paper. The MIC for complete in activation of Staphylococcus Aureus was 7.01 mg Ag/g dry paper for 2:1 ratio and 4.7 mg Ag/g dry paper for 10:1 ratio. The MIC for complete inactivation for Enterococcus Faecalis was 4.01 mg Ag/g dry paper for 2:1 ratio and 4.7 mg Ag/g dry paper for 10:1 ratio. The removal
efficiency for Enterobacter Aerogenes and Klebsialla for all silver contents and both ratios was 100%.

It was observed that for all types of bacteria, the NaBH₄/AgNO₃ ratio of 10:1 resulted in complete inactivation of bacteria in less silver content than the 2:1 ratio and that because the 10:1 ratio resulted in smaller and more

uniform AgNPs which led to more contact between the AgNPs and the bacteria [Ali Bakhtiari-Sardari *et al*., 2020]. These results agree with the findings of the previous studies concerning this subject.

**3.4 Analysis of silver content in The Effluent**

Due to possible human health effects from silver exposure, the silver content in the effluent water was analyzed by AAS. Table 4 represents the relationship between the silver content in the paper and silver release in the effluent.

**Table 4: The relationship between the silver content in the papers and silver in the effluent water.**

| $AgNO_3$ concentration, M | Silver Content in the Effluent, mg/L | |
|---|---|---|
| | 2:1 $NaBH_4/AgNO_3$ ratio | 10:1 $NaBH_4/AgNO_3$ ratio |
| 0.005 | 0 | 0 |
| 0.015 | 0 | 0 |
| 0.03 | 0.021 | 0.043 |
| 0.05 | 0.043 | 0.082 |

As shown in Table 4, the average silver content in the effluent water for the three replicates ranged from 0 to 0.082 mg/L which meets the United States- Environmental Protection Agency (US-EPA) guideline for drinking water of less than 0.1 mg/L [EPA, 2018]. This was due to the stability of silver nanoparticle in the cellulose paper. Sodium borohydride acts not only a reducing agent but also as an ion stabilizer, which prevents silver
ions from aggregation. Moreover, hydroxyl and ether groups in the cellulose fiber play an important role in the stabilization of metal nanoparticles [Ricardo J. *et al.,* 2012]. These results agree with the findings of the previous studies concerning this subject.

**3.5 Turbidity Removal**

Turbidity tests were conducted using turbidity meter, SN 10/1467, Germany, available at the University of
165 Babylon/College of Engineering/ Environmental Engineering Department. Table 5 represents the results obtained before and after filtration through AgNPs papers.

**Table 5: Turbidity results before and after filtration through AgNPs paper for both reduction ratios.**

| AgNO3 concentration, M | 2:1 $NaBH_4/AgNO_3$ ratio | | 10:1 $NaBH_4/AgNO_3$ ratio | |
|---|---|---|---|---|
| | Turbidity before filtering through papers (NTU) | Turbidity after filtering through papers (NTU) | Turbidity before filtering through papers (NTU) | Turbidity after filtering through papers (NTU) |
| 0.005 | 13.5 | 0.92 | 13.5 | 0.82 |
| 0.015 | 12.1 | 0.33 | 12.1 | 0.12 |
| 0.03 | 12.1 | 0.22 | 12.1 | 0.13 |
| 0.05 | 13.5 | 0.87 | 13.5 | 0.81 |

As shown in table 5, the cellulose paper acts as a good point of use filter as all the turbidities were reduced to an acceptable level. Reducing the papers with 10:1 $NaBH_4/AgNO_3$ ratio reduced the turbidity better than 2:1 $NaBH_4/AgNO_3$ ratio.

## 4. Conclusion

Sliver nanoparticles used in this work exhibit a broad size distribution with highly reactive facets. It was observed that chemical reduction of $AgNO_3$ by using $NaBH_4$ as a reducing agent resulted in spherical silver nanoparticles. The ratio of $NaBH_4/AgNO_3$ of 10:1 resulted in smaller sizes of silver nanoparticle and more silver content than the ratio of 2:1 for the same $AgNO_3$ concentration. AgNPs paper provided rapid and effective bactericidal activity as the bacterially contaminated water was filtered through the paper specially Enterococcus Faecalis bacteria which is the first time tested for removal by AgNPs. AgNPs papers can be used a good point of use filters.

## ACKNOWLEDGEMENT

This study was supported by the Department of Environmental Engineering in the University of Babylon. We also appreciate the support of the sanitary lab in the college of Engineering to help to carry out this work.

## ABBREVIATIONS

| | |
|---|---|
| AgNPs | silver nanoparticles |
| $AgNO_3$ | silver nitrate |
| $NaBH_4$ | sodium borohydride |
| SEM | Scanning Electron Microscopy |
| TEM | Transmission Electron Microscopy |
| AAS | Atomic Absorption Spectrometer |
| DPBs | disinfection byproducts |
| NOM | natural organic matter |
| THMs | trihalomethanes |
| AAS | haloacetic acid |
| TDS | Total Dissolved Solids |
| HCL | nitric acid |
| $H_2O_2$ | hydrogen peroxide |
| UTI | Urinary Tract Infections |
| MIC | Minimal Inhibitory Concentration |

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
