# Peer review of "Surface Water Purification using cellulose Paper Impregnated with Silver Nanoparticles"

_Drinking Water Engineering and Science, 2020_

## Referee Comment (RC1) · Anonymous Referee #1 · 8 Sep 2020

This study investigated the surface water purification using cellulose paper impregnated with silver nanoparticles. Generally, this topic is formative. In this study, the material and methods are superficial and experiments should be added. Moreover, the lack of discussion and explanation on the novelty is the major concern for publishing. Therefore, I recommend the authors to emphasize the meaning of this study and to address some issues before its consideration for acceptance. The detailed comments are given:

Introduction 1) Line 44 - "Graphene, activated carbon, and nepheline films have also been studied for AgNP immobilized antibacterial surfaces". What made you choose

for cellulose paper? Suggestion is to address advantages of cellulose paper regarding to other materials. 2) Objective of the research is lacking 3) There has been already done research on silver particles impregnated in the cellulose paper. The novelty of the paper should be explained and added.

Materials and methods 1) Detailed explanation on sampling and water characterisation is missing. Why was this water chosen as a model water? 2) How were the bacteriological analysis done? How were the colonies counted? 3) line 54 – origin of the cellulose paper should be stated. 4) line 56 – why were ratios 2:1 and 10:1 chosen? 5) Figure 1 – should be repeated with different background/ preferably white background 6) Details of the measurements regarding SEM and TEM should be given.

Results 1) Reference on turbidity removal with only cellulose paper should be measured 2) Line 125 Minimum Inhibitory concentration should be mentioned and defined earlier in the manuscript 3) Line 132 – "It was observed that for all types of bacteria, the $NaBH_4/AgNO_3$ ratio of 10:1 resulted in complete inactivation of bacteria in less silver content than the 2:1 ratio and that because the 10:1 ratio resulted in smaller and more uniform AgNPs which led to more contact between the silver nanoparticles and the bacteria." It is difficult to make very clear conclusions if Figures 3 and 4 are compared. Removal efficiencies for figures 3 and 4 are almost the same whereas silver content should be reconsidered in order to draw better conclusions.

Conclusions 1) Line 161 - "AgNPs papers can be used a good point of use filters" – This is strong conclusion since it was not compared to other technologies. 2) I suggest conclusion to be revised.

Abbreviations are not correct and should be corrected.

References Suggestion is to use recent references.

---

## Author Comment (AC1) · 10 Sep 2020

| No. of comment | comment | reviewer | No. of page and line | Response |
|---|---|---|---|---|
| Introduction 1 | What made you choose cellulose paper? | RC1 | Page 2 Line 46 to 50 | done |
| Introduction 2 | Objective of the research is lacking | RC1 | Page 2 Line 51 to 54 | done |
| Introduction 3 | The novelty of the paper should be explained and added. | RC1 | Page 2, line 54 Some new types of bacteria | done |
| Materials and methods 1 | Why was this water chosen as a model water? | RC1 | Page 2 Line 58 t0 59 | done |
| Materials and methods 2 | How were the bacteriological analysis done? | RC1 | Page 4 Line 88 to 96 | mentioned |
| Materials and methods 2 | How were the colonies counted? | RC1 | Page 4 Line 96 to 97 | done |
| Materials and methods 3 | origin of the cellulose paper should be stated. | RC1 | Page 5, line 64 | done |
| Materials and methods 4 | why were ratios 2:1 and 10:1 chosen? | RC1 | Page 4, lines 66 and 67 | done |
| Materials and methods 5 | Figure 1 – should be repeated with different background/ preferably white background | RC1 | | These papers were taken during the study and there is no possibility of repeating them. |
| Results 1 | Reference on turbidity removal with only cellulose paper should be measured | RC1 | | It's measured with only cellulose paper already. |
| Results 2 | Minimum Inhibitory concentration should be mentioned and defined | RC1 | Page 8, lines 137 and 138 | done |

| Results 3 | It is difficult to make very clear conclusions if Figures 3 and 4 are compared. | RC1 | | This concern the silver content concentration, we can conclude the 100% inactivation can be reached with less silver content in 10:1 ratio. |
|---|---|---|---|---|
| Conclusion 1 | "AgNPs papers can be used a good point of use filters" – This is strong conclusion since it was not compared to other technologies | RC1 | | This conclusion was not mentioned as a comparison with other technologies it was based on the results abstained from this study. |
| | Abbreviations are not correct and should be corrected. | RC1 | | done |
| | References Suggestion is to use recent references. | RC1 | | These are the most recent references concerning this stude |

---

## Author Comment (AC2) · 10 Sep 2020

thank you for these notes. Introduction 1)What made you choose cellulose paper? done, Page 2 Line 46 to 50. Introduction 2) Objective of the research is lacking done, Page 2, Line 51 to 54. Introduction 3) The novelty of the paper should be explained and added. done, Page 2, line 54. Some new types of bacteria Materials and methods 1) Why was this water chosen as a model water? done, Page 2, Line 58 t0 59. Materials and methods 2) How were the bacteriological analysis done? mentioned, Page 4, Line 88 to 96. Materials and methods 2) How were the colonies counted? done, Page 4, Line 96 to 97. Materials and methods 3) origin of the cellulose paper should be

stated. done, Page 5, line 64. Materials and methods 4)why were ratios 2:1 and 10:1 chosen? done, Page 4, lines 66 and 67. Materials and methods 5)Figure 1 – should be repeated with different background/ preferably white background. These papers were taken during the study and there is no possibility of repeating them. Results 1)Reference on turbidity removal with only cellulose paper should be measured It's measured with only cellulose paper already. Results 2)Minimum Inhibitory concentration should be mentioned and defined done, Page 8, lines 137 and 138. Results 3) It is difficult to make very clear conclusions if Figures 3 and 4 are compared. This concern the silver content concentration, we can conclude the 100% inactivation can be reached with less silver content in 10:1 ratio. Conclusion 1) "AgNPs papers can be used a good point of use filters" – This is strong conclusion since it was not compared to other technologies. This conclusion was not mentioned as a comparison with other technologies it was based on the results abstained from this study. Abbreviations are not correct and should be corrected. done References Suggestion is to use recent references. These are the most recent references concerning this study

---

## Author Comment (AC3) · 10 Sep 2020

revised manuscript

**Surface Water Purification using cellulose Paper Impregnated with Silver Nanoparticles**

Shahad A. Raheem[1], Alaa H. Alfatlawi[2]

[1] College of Water Resources Engineering/ Al-Qassim Green University, Hilla/Iraq.
5  [2] College of Engineering / University of Babylon, Hilla/Iraq.
*Correspondence to*: Shahad A. Raheem (E.mail:shahad.ak@wrec.uoqasim.edu.iq)

**Abstract.** The objective of this study is to prepare a cellulose paper was impregnated with (AgNPs) for the purpose of water purification (Disinfection (removal of Escherichia C

10  Aureus, Enterococcus Faecalis, Enterobacter Aerogenes, Klebsiella Pneumoniae, and Pro filtration). AgNPs papers were prepared by chemical reduction of silver nitrate (Ag concentrations (0.005 M, 0.015 M, 0.03 M, and 0.05 M) using sodium borohydride (Na agent. Two ratios of $NaBH_4/AgNO_3$ of 2:1 and 10:1 were used to show the effect of reducti and removal efficiencies of AgNPs. AgNPs papers were characterized using Scanning E

15  (SEM) and Transmission Electron Microscopy (TEM). An acid digestion using HCL acid fo the samples in Atomic Absorption Spectrometer (ASS) was conducted to measure the silv AgNPs papers. TEM images showed that the silver nanoparticles size in the papers varies fro Water samples, after filtration through AgNPs papers, were analyzed using (ASS) to concentration in the effluent water. AgNPs paper antibacterial efficiency ranged (99 %

---

## Referee Comment (RC2) · Mona Soliman (Referee) · 2 Dec 2020

Section 2.5, microbial testing: Please describe how the testing of the paper was done. For example: X ml of water was used & the paper was submerged in it for x time. Then sample amount x was extracted ...etc

Section 3, Results and discussion: Placing the study results in wider context remains missing. How does your results compare to previous findings of other studies. Is it same? Is it different? if different can you explain why?

[Figure]

23, 2020.

---

## Author Comment (AC4) · 7 Dec 2020

- Section 2.5, microbial testing: Please describe how the testing of the paper was done? The microbial was conducted for water samples not for AgNPs papers.

- Section 3, Results and discussion: Placing the study results in wider context remains missing. Done

---

## Author Response (AR2)

**Author's response**

- Line 8: probably the word "that" is missing..: "paper that was",

Done
- Line 9: delete "(removal of… mirabilis)" from abstract.

Done
- Line 15: remove "(SEM)" and "(TEM)" from abstract (abbreviations not used anymore in abstract)

Done
- Line 17: varies = varied

Done
- Line 18: (ASS) = ASS

Done
- Line 22: must be "0.1 mg/L. Turbidity".

Done
- Line 25: pathogens are not "breeding" in the water, so do not use that word. At the end of the line: "The"

Done
- Line 31: acid = acids

Done
- Line 32: as mentioned by reviewer "rapidly developing science" should be illustrated by more recent references.

Done
- Use abbreviations, after introduction after first appearance, in the entire document (e.g. now in line 31 AGNPs should be introduced)

Done
- Line 46-54: indicate what is new (in the approach) and contributes to science.

Done

- Line 52: "is" should be "can be", not clear how you can use readily AgNP impregnated paper in water treatment…

Done

- Line 52: rephrase "a study investigates.."

Done

- Rephrase line 58…

Done

- Line 59: Their = Its

Done

- Line 60: bottle = bottles

Done

- Line 69: which were not..

Done

- Line 69: absorbed = adsorbed

Done

- Line 81: the amount of dissolved silver was analysed

Done

- Line 96: was repeated

Done

- Line 113-114, give references of these studies (see comment of reviewer)

Done

- Line 118: use abbreviation here (ASS already introduced in paper)

Done

- Line 124, give references of these studies (see comment of reviewer)

Done

- Line 130-135: the lines between the dots in the Figures 3 and 4 do not have a meaning

Done

- Line 147: use abbreviation for "silver nanoparticles"

Done

- Line 147-148: give references of these studies (see comment of reviewer)

Done

- Line 157: use abbreviation for "silver nanoparticles"

Done

- Line 157-161: give references of these studies (see comment of reviewer)

Done

- Line 172-177: indicate in the conclusion what is a new finding that contributes to science

Done

---

## Author Response (AR3)

**Author's Response**

**- Line 14: introduce ASS**
Done.

**- Line 58: It's = Its**
Done.

**- Check also references: must be [Surname et al., year]**
**Thus:**
**\* Line 113: [Quintero-Quiroz et al., 2019]**
Done.

**\* Line 123-124: [Dankovich and Gray, 2011]**
Done.

**\* Line 161: [Pinto et al., 2012]**
Done.

**- Order references on Surname:**
**\* Line 196: Surname = Quintero-Quiroz**
Done.

**\* Line 223: Surname = Pinto**
Done.